# GAMT: A Geometry-Aware, Multi-view, Training-free Segmentation Framework for Foundation Models in Medical Imaging

Sun Jo[1][0009−0003−5513−6474]*, Ahjin Choi[1][0009−0009−0431−0915]*, and Je Hyeong Hong[1][0000−0003−2797−553X]

Hanyang University, Seoul 04763, Republic of Korea
`{choyw5, caj0328}@hanyang.ac.kr`
*The first two authors share equal contribution.

**Abstract.** Medical image segmentation is a critical task in clinical diagnostics and biomedical research. While deep learning has significantly advanced the field, most existing methods rely on task-specific models that require extensive manual annotations for training or adaptation. Vision foundation models, such as the Segment Anything Model (SAM), offer a promising alternative with their universal segmentation capabilities. However, their application to 3D medical imaging remains limited, especially in zero-shot scenarios involving previously unseen anatomical structures. In this work, we introduce GAMT, a zero-shot, training-free framework that repurposes powerful 2D foundation segmentation models (e.g., SAM, SAM-Med2D) for universal 3D biomedical image segmentation. To bridge the dimensionality gap, GAMT performs slice-wise inference along three orthogonal anatomical planes (axial, coronal, and sagittal) and subsequently fuses the predictions to construct a coherent 3D segmentation mask. Crucially, without any model training or fine-tuning, this framework achieves average Dice Similarity Coefficient (DSC) and Normalized Surface Dice (NSD) scores of dsc-score and nsd-scores, respectively—without requiring model training or fine-tuning. Our code and results are publicly available at https://github.com/SpatialAILab.

**Keywords:** Medical image segmentation · foundation-model · training-free.

## 1  Introduction

Medical image segmentation is a fundamental task in clinical workflows, supporting critical applications such as anatomical structure identification, pathology localization, treatment planning, and longitudinal disease monitoring. Accurate segmentation enables clinicians to extract quantitative biomarkers and visualize target regions with high precision. Traditionally, models such as nnUNet [4] have been widely adopted for 3D medical image segmentation due to their strong performance across various datasets. However, these specialist segmentation models are usually heavily task-specific, often rely on supervised learning with large,

annotated datasets, which are difficult to acquire in the medical domain and they can only segment the data from the same domain as the training set. Recently, the emergence of vision foundation models (VFMs), such as the Segment Anything Model (SAM [7]), has introduced a new paradigm of universal, prompt-driven segmentation. Leveraging such models for medical imaging offers an opportunity to reduce annotation costs and eliminate task-specific training. However, it still encounters some challenges in practical applications – direct application to medical images, particularly volumetric (3D) data, remains challenging. Most VFMs are designed for 2D images, lack geometric understanding of 3D anatomy, and require significant adaptation efforts to perform reliably in the medical domain.

Initial efforts to bridge this gap, such as MedSAM [8] and MedSAM2 [10], have adapted VFMs to the medical domain by fine-tuning them on curated 2D and 3D medical datasets respectively. While improving performance on specific tasks, these methods sacrifice the zero-shot generalization ability of the original models by re-introducing training requirements. More recent works have focused on interactive 3D segmentation; for instance, SAM-Med3D [11] and SegVol [2] employ prompt propagation strategies to extend 2D inference across volumetric slices. s. Similarly, VISTA3D [5] and nnInteractive [3] explore multi-view consistency and 3D-aware attention mechanisms to improve segmentation robustness without requiring full supervision. Despite these innovations, they often still depend on partial training, exhibit limited 3D consistency, or incur substantial computational overhead, restricting their utility in truly zero-shot, training-free clinical scenarios where models must robustly segment novel structures unseen during development.

In this context, the "CVPR 2025 Foundation Models for Interactive 3D Biomedical Image Segmentation Challenge" presents the first large-scale benchmark for evaluating foundation models in interactive 3D medical image segmentation. The competition provides over 200,000 3D image–mask pairs across diverse anatomical structures and imaging modalities. Participants are required to develop interaction-efficient segmentation frameworks that can iteratively refine predictions based on user prompts (points).

This challenge emphasizes three core aspects: (1) enabling interactivity in 3D segmentation using foundation models, (2) handling diverse biomedical image types (CT, MRI, PET, Microscopy and Ultrasound) with high-quality annotations, and (3) jointly optimizing segmentation accuracy and computational efficiency. The top-performing algorithms will also be considered for integration into 3D Slicer to promote clinical usability and annotation efficiency in real-world scenarios.

Despite recent advances, foundation models for medical segmentation still face two critical hurdles for practical deployment: the reliance on costly training or fine-tuning, and the geometric inconsistencies arising from single-view (e.g., axial) inference on inherently 3D data. These challenges hinder their application in zero-shot clinical settings where adaptability and accuracy are paramount.

To overcome these limitations, we introduce **GAMT**—a **G**eometry-**A**ware, **M**ulti-view, **T**raining-free segmentation framework designed for 3D biomedical image segmentation using 2D foundation models. Our key motivation is to fully exploit the generalization ability of pre-trained models like SAM[7] and SAM-Med2D[1] without requiring any model updates, while enhancing 3D spatial consistency through geometry-aware propagation and fusion.

GAMT is composed of the following core components:

– **Three-view Prompt Inference:** We perform slice-wise inference using point and box prompts along the *axial*, *coronal*, and *sagittal* directions. Each view provides complementary spatial information that helps overcome the limitations of single-plane inference.
– **Prompt Propagation Strategy:** To ensure intra-view consistency, we implement a sequential prompt refinement mechanism that propagates segmentation guidance from one slice to the next using distance-based keypoint selection and clustering strategies.
– **Geometry-aware 3D Fusion:** After multi-view inference, the resulting masks are aggregated via a voting-based fusion strategy followed by lightweight 3D post-processing (connected component filtering and smoothing) to generate a clean and anatomically consistent 3D prediction.
– **Training-free and Model-agnostic Design:** Our method is entirely training-free and agnostic to the foundation model backbone, making it universally applicable to any SAM-compatible model.

We evaluate GAMT on the "CVPR 2025 Foundation Models for Interactive 3D Biomedical Image Segmentation Challenge" dataset, achieving competitive Dice Similarity Coefficient (DSC) and Normalized Surface Dice (NSD) across multiple anatomical structures **without any model retraining or fine-tuning**.

Our main contributions can be summarized as follows:

– We propose **GAMT**, a universal training-free framework for 3D medical image segmentation using 2D foundation models.
– We design a multi-view prompting and fusion strategy that improves geometric consistency and volumetric coverage.
– We demonstrate the effectiveness of GAMT across diverse anatomical targets, achieving competitive accuracy with minimal computational cost.

## 2   Method

### 2.1   Preprocessing

We follow a preprocessing strategy similar to SAM-Med3D [11], tailored for training-free inference with 2D foundation models on 3D volumes. Our preprocessing pipeline commences by normalizing voxel intensities via percentile-based range adjustment over the non-background foreground region. The voxel grid

is then reconstructed to reflect its true physical scale by applying the provided spacing keys. This process ensures that each axis is metrically accurate and represents the original anatomical dimensions. A key step for efficiency is foreground-centered cropping, where we extract a region of interest around the target anatomy (default: $256 \times 256 \times 256$) to significantly reduce the computational domain. From this processed 3D volume, we generate 2D slices along three orthogonal views (axial, coronal, and sagittal) to capture complementary spatial features. Finally, all geometric transformations, such as cropping coordinates and spacing, are meticulously tracked to enable accurate mapping of the 2D predictions back to the original 3D space.

This preprocessing pipeline enables efficient and anatomically faithful slice-wise segmentation using 2D SAM-based foundation models, while preserving geometric consistency and reducing computational burden.

## 2.2 Proposed Method

We propose **GAMT**—a **G**eometry-**A**ware, **M**ulti-view, **T**raining-free segmentation framework for 3D medical image segmentation. Our method utilizes 2D foundation models in a prompt-guided, slice-wise fashion across three anatomical planes: axial, coronal, and sagittal. The GAMT architecture is illustrated in Figure 1 and the inference pipeline is summarized in Figure 2.

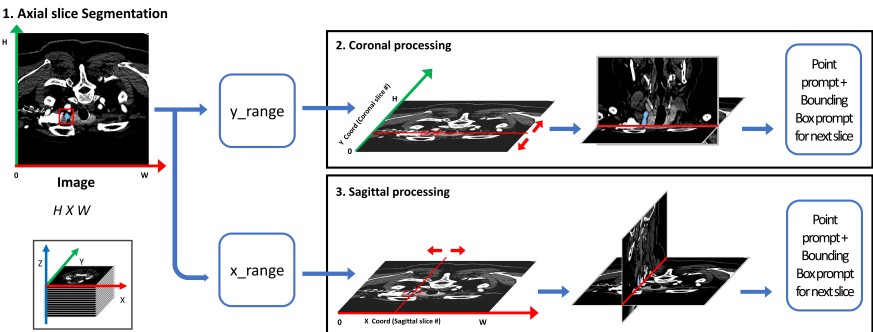

**Fig. 1.** Overview of the GAMT Multi-View Inference and Propagation Strategy.

**GAMT: Multi-View Inference and Propagation Strategy** Our core mechanism for achieving robust 3D segmentation is a multi-view inference and propagation strategy, which is depicted in Figure 1. This strategy ensures 3D consistency by dynamically generating and refining prompts while propagating information across and within three orthogonal planes.

**1. Primary Axis (Axial) Segmentation and Range Determination** The process is initiated on a primary axial slice using an initial bounding box. Once the segmentation mask for this slice is generated, it serves as a spatial reference for the other views. To determine the relevant processing scope, we analyze the mask's projection onto the Y and X axes. Specifically, we compute the sum of segmented pixels along each axis and identify a threshold corresponding to a high percentile (e.g., the 90th percentile) of these sums. The ranges of coordinates that exceed this threshold are then designated as the regions of interest for processing in the coronal and sagittal views, respectively. This data-driven approach ensures that subsequent processing is focused only on slices containing significant portions of the target anatomy.

**2. Orthogonal View Processing and Intra-View Propagation** Next, the framework performs segmentation on the two orthogonal views, guided by the ranges determined in the previous step. A key to our method is the intra-view prompt propagation mechanism employed within both the coronal and sagittal passes.

**Coronal & Sagittal Processing:** The y_range guides the selection of coronal slices for processing. For each selected slice, the framework performs segmentation. The novelty lies in how the output of one slice informs the next.

**Automatic Prompt Refinement:** For each segmented 2D slice within a given view (e.g., coronal), we generate a new set of refined prompts to guide the segmentation of the subsequent slice. This involves two steps:

- **Bounding Box Update:** A new, tight-fitting bounding box is extracted directly from the current slice's segmentation mask. This allows the prompt to adapt to the changing cross-section of the anatomy.
- **Internal Point Prompt Generation:** To enhance robustness, a new positive point prompt is automatically determined. We first apply a distance transform to the mask to identify pixels farthest from the boundary, which typically correspond to the object's core. We then analyze the intensity distribution of the original image within this core region. Finally, a point with both a high distance transform value and a representative intensity is selected as the new point prompt. This dual-criteria approach ensures the point remains stable and centered within the target structure.

An analogous process, guided by the x_range, is performed independently in the sagittal view. By continuously refining and propagating prompts within each view, our method robustly tracks the anatomical structure slice-by-slice.

**3. Volumetric Consistency Across Views** This entire multi-view propagation strategy allows GAMT to leverage complementary geometric information, correct errors from a single-view perspective, and maintain volumetric coherence throughout the segmentation process.

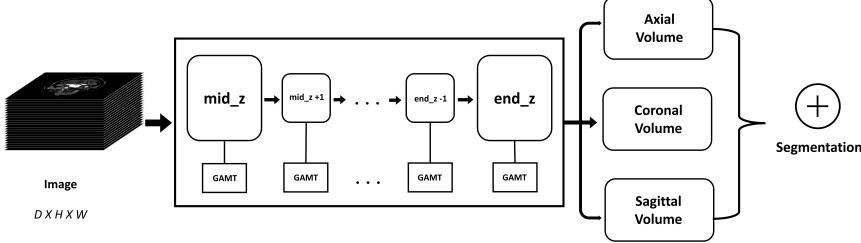

**Fig. 2.** The End-to-End Inference Pipeline of GAMT

**The End-to-End Inference Pipeline of the GAMT Framework** Figure 2 provides a high-level schematic of our entire inference process. Given a 3D input volume (D×H×W) as input, our framework processes it sequentially along a primary axis. (Slice-wise Propagation) Starting from a central anchor slice (mid_z), the core GAMT module is applied iteratively to each subsequent slice until the end of the target region (end_z). As detailed in Fig. 1, this GAMT module performs its own internal multi-view analysis at each step. This iterative process results in the generation of three distinct volumetric masks, one for each orthogonal plane (Axial, Coronal, and Sagittal). Finally, these three volumes are aggregated and fused to produce the single, coherent final segmentation.

### 2.3   Post-processing

After multi-view segmentation is completed and fused, we perform a series of post-processing steps to restore the segmentation mask to its original coordinate space and improve anatomical consistency. This includes uncropping, resampling, and 3D volume refinement.

**Uncropping.** As the inference is performed on cropped regions-of-interest (ROIs), the predicted mask is first uncropped back to the original image size. The offset and crop dimensions used during preprocessing are stored in data keys and used to accurately reposition the predicted region within the full 3D volume.

**Resampling to Original Spacing.** To align the output with the original input file, the processed segmentation mask is resized back to the original image's raw array dimensions. This inverse-resampling step is performed using nearest-neighbor interpolation, ensuring a direct voxel-to-voxel correspondence between the final mask and the initial, unprocessed data grid.

**Smoothing and Connected Component Filtering.** To remove noise and non-anatomical artifacts, we apply connected component analysis in 3D. Only

the largest component is preserved per structure. Optionally, morphological operations such as binary opening or closing are used to smooth rough mask boundaries and reduce voxel-level artifacts introduced during fusion.

This lightweight post-processing pipeline ensures that the final segmentation results are anatomically clean, spatially accurate, and suitable for downstream medical analysis or evaluation.

## 3    Experiments

### 3.1    Dataset and evaluation metrics

The development set is an extension of the CVPR 2024 MedSAM on Laptop Challenge [9], including more 3D cases from public datasets[1] and covering commonly used 3D modalities, such as Computed Tomography (CT), Magnetic Resonance Imaging (MRI), Positron Emission Tomography (PET), Ultrasound, and Microscopy images. The hidden testing set is created by a community effort where all the cases are unpublished. The annotations are either provided by the data contributors or annotated by the challenge organizer with 3D Slicer [6] and MedSAM2 [10]. In addition to using all training cases, the challenge contains a coreset track, where participants can select 10% of the total training cases for model development.

For each iterative segmentation, the evaluation metrics include Dice Similarity Coefficient (DSC) and Normalized Surface Distance (NSD) to evaluate the segmentation region overlap and boundary distance, respectively. The final metrics used for the ranking are:

- DSC_AUC and NSD_AUC Scores: AUC (Area Under the Curve) for DSC and NSD is used to measure cumulative improvement with interactions. The AUC quantifies the cumulative performance improvement over the five click predictions, providing a holistic view of the segmentation refinement process. It is computed only over the click predictions without considering the initial bounding box prediction as it is optional.
- Final DSC and NSD Scores after all refinements, indicating the model's final segmentation performance.

In addition, the algorithm runtime will be limited to 90 seconds per class. Exceeding this limit will lead to all DSC and NSD metrics being set to 0 for that test case.

### 3.2    Implementation details

**Environment settings** The development environments and requirements are presented in Table 1.

---

[1] A complete list is available at https://medsam-datasetlist.github.io/

**Table 1.** Development environments and requirements.

| System | Ubuntu 20.04.6 LTS |
|---|---|
| CPU | AMD Ryzen 9 5900X 12-Core Processor |
| RAM | 3×32GB; 3.2MT/s (DDR4-3200) |
| GPU (number and type) | One NVIDIA GeForce RTX 3090 Ti 16G |
| CUDA version | 10.1 |
| Programming language | Python 3.10.8 |
| Deep learning framework | e.g., torch 1.13.1, torchvision 0.14.1 |

**Inference Configuration** The challenge is structured into two distinct tracks: one utilizing the complete training set and another using a 10% coreset. As our proposed framework, GAMT, is entirely training-free, this distinction did not pertain to model training. Instead, we adapted our inference strategy and choice of foundation model for each track to adhere to the competition's computational time limits.

For the complete training set track, which involves processing a larger number of test cases, we employed a streamlined, single-view inference strategy using only the primary axial view. The foundation model backbone for this track was the pre-trained SAM with a ViT-L encoder. For the 10% coreset track, we performed inference on both the axial and coronal planes and fused the results, utilizing the pre-trained SAM-Med2D model with a ViT-B encoder. Furthermore, to ensure compliance with the challenge's computational time limit of 90 seconds per class, we limited the interactive refinement process to a single iteration for all cases in both tracks.

## 4   Results and discussion

### 4.1   Quantitative results on validation set

We present the quantitative evaluation of our GAMT framework on the validation set, with results analyzed separately for the all-data and coreset tracks as shown in Table  3 and Table  2, respectively. Our approach establishes a strong performance baseline for fully training-free, zero-shot 3D segmentation.

On the all-data track, where GAMT employed a streamlined single-view (axial-only) strategy with a SAM ViT-L backbone, the framework demonstrated its capability as a robust generalist model. As shown in Table  3, GAMT achieved its most competitive results in the PET modality, recording a Final NSD of 0.5836. In other modalities like CT and Microscopy, it provided foundational performance with Final NSD scores of 0.5359 and 0.4812, respectively. However, the single-view approach showed clear limitations on Ultrasound data (Final NSD 0.0962), indicating that this modality may require more complex spatial information for accurate segmentation.

For the coreset track, we utilized a more computationally intensive dual-view (axial and coronal) approach with the domain-adapted SAM-Med2D backbone.

**Table 2.** Quantitative evaluation results of the validation set on the **coreset track**.

| Modality | Methods | DSC AUC | NSD AUC | DSC Final | NSD Final |
|---|---|---|---|---|---|
| CT | SAM-Med3D | 2.2408 | 2.2213 | 0.5590 | 0.5558 |
| | VISTA3D | 3.1689 | 3.2652 | 0.8041 | 0.8344 |
| | SegVol | 2.9809 | 3.1235 | 0.7452 | 0.7809 |
| | nnInteractive | 3.4337 | 3.5743 | 0.8764 | 0.9165 |
| | GAMT | 1.7571 | 1.8240 | 0.4404 | 0.4572 |
| MRI | SAM-Med3D | 1.5222 | 1.5226 | 0.3903 | 0.3964 |
| | VISTA3D | 2.5895 | 2.9683 | 0.6545 | 0.7493 |
| | SegVol | 2.6719 | 3.1535 | 0.6680 | 0.7884 |
| | nnInteractive | 2.6975 | 3.0292 | 0.7302 | 0.8227 |
| | GAMT | 1.2063 | 1.4595 | 0.3046 | 0.3685 |
| Microscopy | SAM-Med3D | 0.1163 | 0 | 0.0291 | 0 |
| | VISTA3D | 2.1196 | 3.2259 | 0.5478 | 0.8243 |
| | SegVol | 1.6846 | 2.9716 | 0.4211 | 0.7429 |
| | nnInteractive | 2.3311 | 3.1109 | 0.5943 | 0.7890 |
| | GAMT | 0.9727 | 1.6044 | 0.2432 | 0.4011 |
| PET | SAM-Med3D | 2.1304 | 1.7250 | 0.5344 | 0.4560 |
| | VISTA3D | 2.6398 | 2.3998 | 0.6779 | 0.6227 |
| | SegVol | 2.9683 | 2.8563 | 0.7421 | 0.7141 |
| | nnInteractive | 3.1877 | 3.0722 | 0.8156 | 0.7915 |
| | GAMT | 1.8659 | 1.5855 | 0.4665 | 0.3964 |
| Ultrasound | SAM-Med3D | 1.4347 | 1.9176 | 0.4102 | 0.5435 |
| | VISTA3D | 2.8655 | 2.8441 | 0.8105 | 0.8079 |
| | SegVol | 1.2438 | 1.8045 | 0.3109 | 0.4511 |
| | nnInteractive | 3.3481 | 3.3236 | 0.8547 | 0.8494 |
| | GAMT | 1.4336 | 1.5338 | 0.3584 | 0.3835 |

**Table 3.** Quantitative evaluation results of the validation set on the **all-data track**.

| Modality | Methods | DSC AUC | NSD AUC | DSC Final | NSD Final |
|---|---|---|---|---|---|
| CT | SAM-Med3D | 2.2408 | 2.2213 | 0.5590 | 0.5558 |
| | VISTA3D | 3.1689 | 3.2652 | 0.8041 | 0.8344 |
| | SegVol | 2.9809 | 3.1235 | 0.7452 | 0.7809 |
| | nnInteractive | 3.4337 | 3.5743 | 0.8764 | 0.9165 |
| | GAMT | 1.8974 | 2.1438 | 0.4744 | 0.5359 |
| MRI | SAM-Med3D | 1.5222 | 1.5226 | 0.3903 | 0.3964 |
| | VISTA3D | 2.5895 | 2.9683 | 0.6545 | 0.7493 |
| | SegVol | 2.6719 | 3.1535 | 0.6680 | 0.7884 |
| | nnInteractive | 2.6975 | 3.0292 | 0.7302 | 0.8227 |
| | GAMT | 1.3828 | 1.6267 | 0.3457 | 0.4067 |
| Microscopy | SAM-Med3D | 0.1163 | 0 | 0.0291 | 0 |
| | VISTA3D | 2.1196 | 3.2259 | 0.5478 | 0.8243 |
| | SegVol | 1.6846 | 2.9716 | 0.4211 | 0.7429 |
| | nnInteractive | 2.3311 | 3.1109 | 0.5943 | 0.7890 |
| | GAMT | 1.0882 | 1.9249 | 0.2720 | 0.4812 |
| PET | SAM-Med3D | 2.1304 | 1.7250 | 0.5344 | 0.4560 |
| | VISTA3D | 2.6398 | 2.3998 | 0.6779 | 0.6227 |
| | SegVol | 2.9683 | 2.8563 | 0.7421 | 0.7141 |
| | nnInteractive | 3.1877 | 3.0722 | 0.8156 | 0.7915 |
| | GAMT | 2.5405 | 2.3344 | 0.6351 | 0.5836 |
| Ultrasound | SAM-Med3D | 1.4347 | 1.9176 | 0.4102 | 0.5435 |
| | VISTA3D | 2.8655 | 2.8441 | 0.8105 | 0.8079 |
| | SegVol | 1.2438 | 1.8045 | 0.3109 | 0.4511 |
| | nnInteractive | 3.3481 | 3.3236 | 0.8547 | 0.8494 |
| | GAMT | 0.6131 | 0.3847 | 0.1533 | 0.0962 |

The quantitative results in Table 2 provide a valuable benchmark for this zero-shot, multi-view strategy. In this setting, GAMT achieved Final NSD scores of 0.4572 on CT, 0.3685 on MRI, and 0.3835 on Ultrasound data. While these results did not consistently surpass those of specialized or training-based methods like nnInteractive, they underscore the performance that can be achieved without any training or fine-tuning on the target dataset. This highlights both the potential and the current limitations of a purely off-the-shelf, multi-view fusion approach.

### 4.2   Qualitative results on validation set

Figure 3 contains examples of good segmentation results on CT LungMask, MR Chaos, PET autoPET and MR_ISLES2022 data. The corresponding DSC and NSD scores were 97.13% and 86.42% for CT LungMask, 84.56% and 82.33% for MR Chaos, 83.44% and 56.18% for PET autoPET,and 84.56% and 100% for MR_ISLES2022. Figure 4 depicts examples with bad segmentation results. The observed failure cases in segmentation can be primarily attributed to the method's fundamental limitation—its high dependency on the type and quality of the initial prompt. We provide a comprehensive analysis of this limitation in Section 4.4.

### 4.3   Results on final testing set

This is a placeholder. No need to show testing results now. We will announce the testing results during CVPR (6.11) then you can add them during the revision phase.

### 4.4   Limitation and future work

**Limitations** The primary limitation of GAMT is its high dependency on the type and quality of the initial prompt. Our method's propagation architecture is fundamentally designed to commence from a well-defined spatial region provided by an initial bounding box on a central anchor slice (mid_ z). Consequently, the quality of the entire 3D segmentation is highly contingent on this initial box prompt. If the initial box fails to capture the target structure accurately, these errors can propagate throughout the volume, leading to a suboptimal segmentation. This design choice leads to a significant performance degradation in scenarios where the initial prompt is not a bounding box. For instance, in cases such as vessel segmentation where only point prompt is provided initially, our framework struggles to establish a stable starting region for propagation. The current implementation lacks a robust mechanism to translate a sparse point prompt into a reliable initial segmentation mask, making it ill-suited for such use cases without modification. Furthermore, the performance can degrade in scenarios with low contrast, where the intensity distribution of the target anatomy is not clearly distinguishable from the surrounding background tissue. This is because our internal point prompt generation mechanism for slice-to-slice propagation relies on intensity profiles to identify the core of the structure.

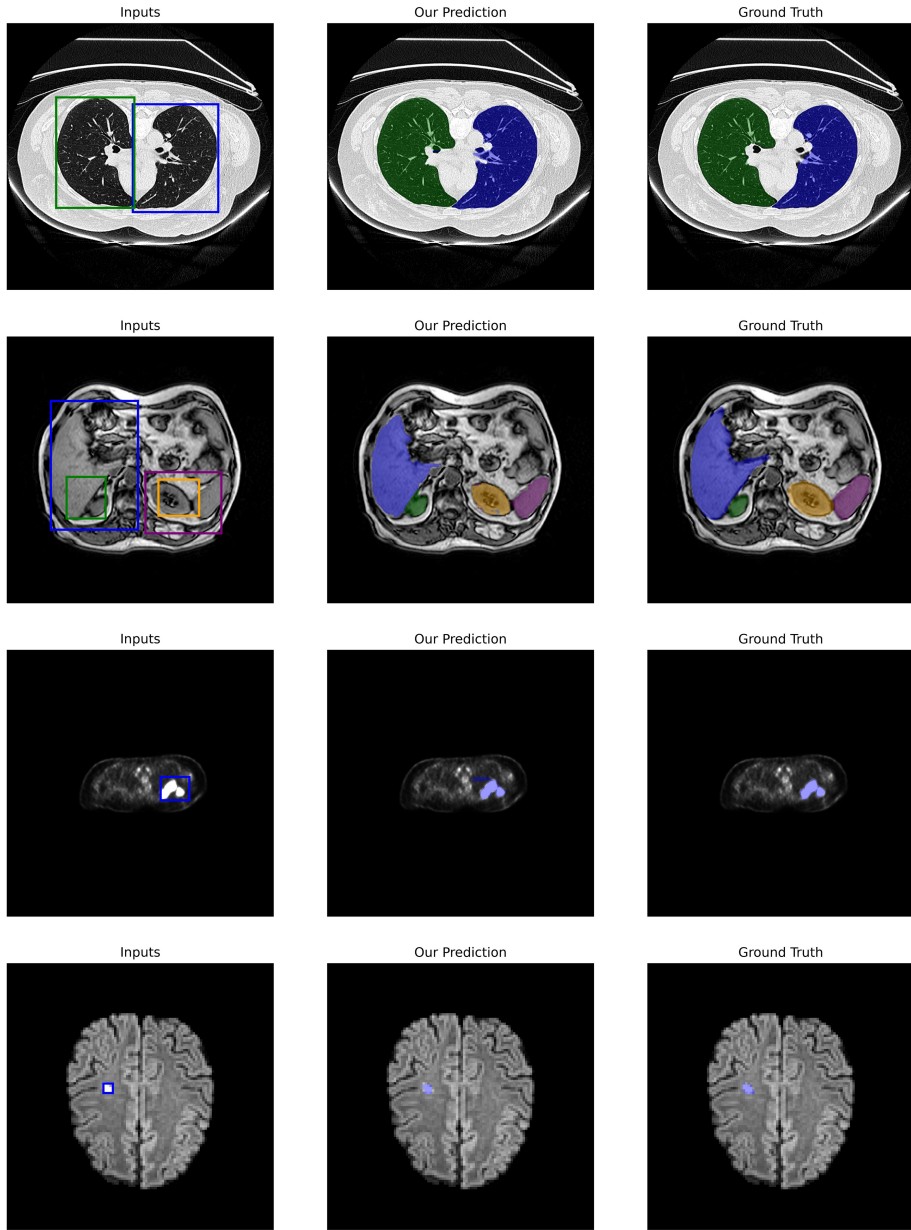

**Fig. 3.** Examples of Good Segmentation Results: The first row contains a CT Lung-Mask data, the second row is a MR Chaos data, the third row is PET autoPET data and the last row is a MR_ISLES2022 data.

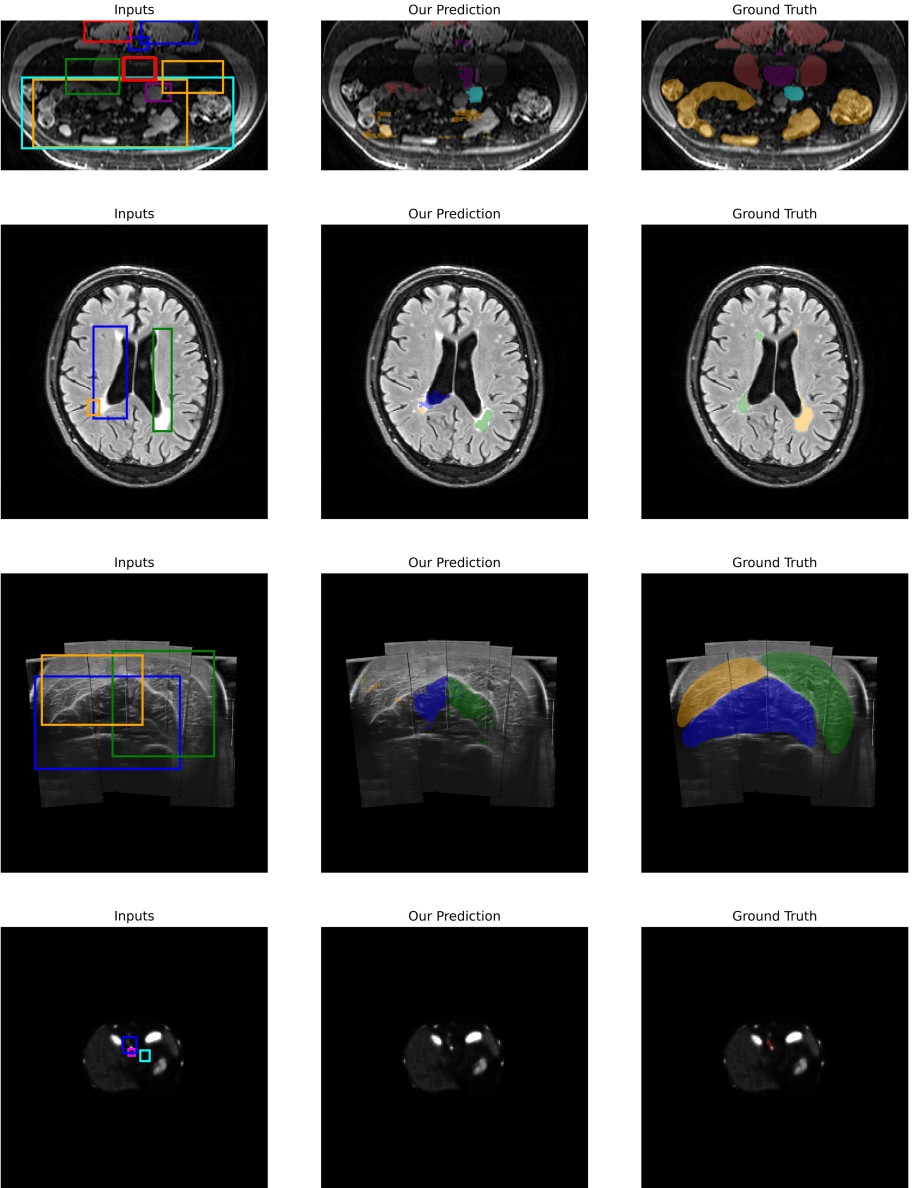

**Fig. 4.** Examples of Bad Segmentation Results: The first row contains a MR_totalseg data, the second row is an MR_WMH_FLAIR_Singapore data, the third row is US_Low-limb-Leg35 data and the last row is a PET_autoPET data.

**Future Work** Building on the strengths of our training-free, generalist approach, our future work will focus on extending GAMT's application to anatomies that are notoriously difficult for traditional supervised models. A primary area of investigation will be the segmentation of highly complex and variable structures, such as vasculature (vessels). These structures often suffer from a lack of large, annotated training datasets and exhibit significant morphological variations between patients, making them ideal candidates for a zero-shot framework like GAMT. We plan to leverage our method's ability to operate without prior training to provide robust segmentation for these novel structures, thereby addressing a significant challenge in medical imaging analysis.

## 5   Conclusion

In this work, we present GAMT, a novel, training-free framework designed to adapt powerful, pre-trained 2D foundation models for 3D biomedical image segmentation. Our method introduces a multi-view propagation strategy, performing slice-by-slice inference along orthogonal axial and coronal planes. The core of our approach is an automatic prompt refinement mechanism that ensures spatial consistency by dynamically updating bounding box and point prompts between adjacent slices. By fusing the segmentation results from these complementary views, GAMT achieves competitive segmentation accuracy on a diverse range of anatomical structures. Most notably, these results are obtained without any task-specific training or fine-tuning, demonstrating a practical and efficient path toward leveraging foundation models in zero-shot clinical settings. We have made our code and results publicly available on GitHub to encourage further research and collaboration within the community.

**Acknowledgements** We thank all the data owners for making the medical images publicly available and CodaLab [12] for hosting the challenge platform. This study was funded by X (grant number X).

**Disclosure of Interests.** The authors have no competing interests to declare that are relevant to the content of this article.

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

**Table 4.** Checklist Table. Please fill out this checklist table in the answer column. (**Delete this Table in the camera-ready submission**)

| Requirements | Answer |
|---|---|
| A meaningful title | Yes/No |
| The number of authors ($\leq 6$) | Number |
| Author affiliations and ORCID | Yes/No |
| Corresponding author email is presented | Yes/No |
| Validation scores are presented in the abstract | Yes/No |
| Introduction includes at least three parts: background, related work, and motivation | Yes/No |
| A pipeline/network figure is provided | Figure number |
| Pre-processing | Page number |
| Strategies to data augmentation | Page number |
| Strategies to improve model inference | Page number |
| Post-processing | Page number |
| Environment setting table is provided | Table number |
| Training protocol table is provided | Table number |
| Ablation study | Page number |
| Efficiency evaluation results are provided | Table number |
| Visualized segmentation example is provided | Figure number |
| Limitation and future work are presented | Yes/No |
| Reference format is consistent. | Yes/No |
| Main text $>= 8$ pages (not include references and appendix) | Yes/No |