# OpenReview forum: "GAMT: A Geometry-Aware, Multi-view, Training-free Segmentation Framework for Foundation Models in Medical Imaging"
_thecvf.com/CVPR/2025/Workshop/MedSegFM — CVPR 2025 Workshop MedSegFM Submission_

### Official Review · Reviewer_s2nj · 2025-09-08
**In this paper, the authors propose a framework that uses pretrained 2D medSAM models for 3D medical segmentation without fine-tuning or model training. The method combines segmentation mask from three anatomical planes to construct a coherent 3D segmentation mask.**

**Rating:** 6
**Confidence:** 4

**Review:**

The advantage of this framework is that it is zero-shot, which means any training or fine-tuning is not needed. It is basically operated on the inference stage and with some post-processing to refine the final result.

However, the framework also has some drawbacks, and there are still something needing to be enriched:
1. In the Primary Axis (Axial) Segmentation and Range Determination section, the method sets a threshold (e.g., the 90th percentile of sums of segmented pixels along each axis). I am wondering if this percentage is obtained by doing some ablation studies or if it is just an empirical value. Furthermore, is it best that this range is fixed for all cases? Maybe someone needs a higher percentage or a lower percentage to get better performance.
2. Based on my understanding, for each slice along the z direction, a 3D segmentation mask for x_range and y_range should be generated. In this way, I think it is very repetitive, as the overlap should be very large for the  x_range and y_range results of each slice, and maybe only one slice with the largest mask area is needed; then get the 3D segmentation mask along the x and y axes, respectively.
3. Better to add an up arrow for each evaluation metric in the tables.
4. From the performance tables, it seems that the current GAMT framework struggles with competing with the baseline methods, which might also indicate the limitation of using 2D segmentation models for the 3D segmentation and the high requirement for the initial accurate prompts. Likewise, it is mentioned that the current method can't work well with the cases without an initial box prompt, like vessel cases.

---

> ### Author Rebuttal · Authors · 2025-11-05
>
> We sincerely appreciate the reviewer’s thoughtful and detailed comments, which have greatly contributed to enhancing the quality and presentation of our manuscript. We have carefully addressed these suggestions and made corresponding revisions in the manuscript.
>
> 1. Methodological Justifications and Sensitivity Analysis
> This threshold is rather a practical mechanism for defining the processing range to maintain efficiency under the challenge’s runtime limit. We conducted a sensitivity analysis testing 80%, 90%, and 95% thresholds. The analysis showed minimal performance differences between 90%, and 95% thresholds. However, a lower threshold (80%) significantly increased the number of slices processed and the overall inference time, which exceeded the 90-second limit imposed by the challenge. Therefore, the 90% threshold was selected as the most balanced configuration between robust performance and computational feasibility.

---

### Official Review · Reviewer_vhmt · 2025-09-30
**Necessity of Ablation, Baseline Comparison, and Broader Evaluation**

**Rating:** 6
**Confidence:** 4

**Review:**

This paper introduces a training-free universal segmentation framework, GAMT, which incorporates a multi-view prompting and fusion strategy and leverages a 2D foundation model for 3D image analysis.
However, several concerns arise:
1. Ablation studies should be included to validate the effectiveness of the proposed modules.
2. Although the method is training-free, it does not achieve performance comparable to competing baselines. The advantages of GAMT, particularly in relation to SAM-Med3D, should be explicitly clarified and emphasized.
3. The authors are encouraged to provide the performance results of the coreset track participants for a more comprehensive comparison.

---

> ### Author Rebuttal · Authors · 2025-11-05
>
> We sincerely appreciate the reviewer’s thoughtful and detailed comments, which have greatly contributed to enhancing the quality and presentation of our manuscript. We have carefully addressed these suggestions and made corresponding revisions in the manuscript.
>
> 1. View Setting
> The systematic improvement in accuracy from Single-view to Multi-view confirms that incorporating orthogonal views addresses 3D geometric inconsistencies. However, the definitive challenge submission utilized Dual-view, despite Multi-view being superior, to comply with the strict 90-second time limit. This challenge time limit also strategically limited the processed slices (90% threshold).
>
> 2. Fusion
> The choice of a voting-based fusion was intentional to address the heterogeneous reliability across the three orthogonal views. The axial view, guided by the initial user prompt, typically provides the most reliable segmentation, whereas the coronal and sagittal views may exhibit orientation-specific artifacts. Simple fusion methods such as logical OR or averaging may therefore amplify noise and false positives.

---

### Comment · Reviewer_XyAr · 2025-10-13
**Meta-Review: Major Revision required due to lack of Ablation Studies and Methodological justifications**

While the proposed "training-free" framework addresses a significant practical need in medical imaging, the current manuscript holds critical deficiencies in experimental validation and performance analysis that must be addressed. A Major Revision is recommended based on the consolidated feedback from all reviewers.
The authors must address the following key issues:
1. Critical Lack of Ablation Studies
The manuscript proposes several core components (multi-view inference, prompt propagation, geometry-aware fusion) without validating them independently. Readers cannot determine the source of any performance gains or the necessity of the framework's complexity.
Action Required: Include ablation experiments comparing:
View settings: Single-view (axial only) vs. Dual-view vs. Three-view.
Propagation: With vs. without the proposed automatic prompt refinement/propagation strategy.
Fusion: The proposed voting-based fusion vs. simple baseline fusion methods (e.g., logical OR, averaging).
2. Performance Analysis and Positioning
The quantitative results (Tables 2 & 3) show GAMT underperforming compared to baselines. While "training-free" is an advantage, the trade-off between zero training cost and lower performance is not adequately discussed.
Action Required:
Reframe the contribution to emphasize it as a strong zero-shot baseline rather than a performance leader. Discuss the performance/cost trade-off elegantly.
Explicitly clarify the advantage of GAMT over simply applying SAM-Med3D directly.
Provide an efficiency analysis (inference time, GPU memory) compared to training-based methods to highlight the practical benefits.
If possible, include full results from the challenge's coreset track for comprehensive comparison.
3. Methodological Justifications and Sensitivity Analysis
Several design choices appear empirical and lack justification.
Action Required:
Thresholding: The 90th percentile threshold for range determination appears arbitrary. Conduct a sensitivity analysis (e.g., testing 80%, 95%) to justify this selection.
Initial Prompt Dependency: Quantify the framework's sensitivity to the quality of the initial bounding box (e.g., by perturbing the initial box).
Clarity: Clarify the description of generating 3D masks for x_range and y_range  along the z-axis, which currently reads as potentially redundant or inefficient.
4. Presentation Details
Add arrows (e.g., DSC ↑, NSD ↑) to tables for clarity.

---

> ### Author Rebuttal · Authors · 2025-11-05
>
> We appreciate the reviewer's constructive feedback on the performance positioning and the necessity of discussing the trade-off. We agree that our contribution must be reframed to emphasize GAMT's strength as a robust zero-shot baseline rather than a performance leader.
>
> Novelty and Core Contribution
>
> Our primary contribution lies in exploring a new, highly practical research paradigm: investigating the capabilities of off-the-shelf 2D Foundation Models (VFMs) to accomplish 3D medical segmentation tasks. GAMT is a novel, training-free framework that achieves competitive results with zero training cost.
>
> A key contribution of GAMT is its inherent robustness to novel complex anatomies, unlike training-based models.
>
> •	Robustness to Morphological Variation: Training-based models, such as SAM-Med3D, are optimized on a curated dataset.
> Their performance can severely degrade when segmenting anatomies with high morphological complexity and variability—such as vasculature (arteries and veins).
>
> •	Input-Driven Segmentation: GAMT, conversely, operates purely on an inference stage framework by dynamically generating and refining prompts based on the input volume's internal features and geometry. This means the segmentation is guided by the instance itself, not by prior learned representations.
>
> To contextualize our results and address the performance comparison, we summarize our key insights:
> •	Zero-Shot Baseline Positioning: GAMT is explicitly optimized for cost-efficiency and immediate generalization. We will reframe our contribution as serving as the strongest practical zero-shot baseline in the domain, a critical solution for scenarios where data and resources are scarce.
>
> •	The Efficiency-Performance Trade-Off: The observed performance difference from training-based SOTA methods (like nnInteractive or VISTA3D) is a conscious and necessary trade-off.
>
> o	Constraint Impact: The challenge's strict 90-second runtime limit and the forced restriction to a single interactive refinement fundamentally hampered GAMT's performance.
>
> o	Data Diversity Constraint: The challenge dataset's extremely high diversity (covering CT, MRI, PET, Ultrasound, and Microscopy, with over 200,000 3D image-mask pairs) is crucial for generalist models. However, our training-free framework and the reliance on an off-the-shelf VFM meant we had limitations in extracting the ideal initial keypoint prompt across all diverse modalities and structures in a truly robust manner.

---

### Decision · Program_Chairs · 2025-11-12

Accept